# Pharmacological Modulation of Excitotoxicity through the Combined Use of NMDA Receptor Inhibition and Group III mGlu Activation Reduces TMT-Induced Neurodegeneration in the Rat Hippocampus

**DOI:** 10.3390/ijms24098249

**Published:** 2023-05-04

**Authors:** Ekaterina V. Pershina, Irina Yu. Chernomorets, Dmitry A. Fedorov, Vladimir I. Arkhipov

**Affiliations:** Institute of Theoretical and Experimental Biophysics, Russian Academy of Sciences, 142290 Pushchino, Moscow Region, Russia

**Keywords:** neurodegeneration, hippocampus, excitotoxicity, neuroinflammation, trimethyltin (TMT), NMDArs, metabotropic glutamate receptors (mGluRs), memantine, VU0422288, astrocytes, microglia

## Abstract

We studied the neuroprotective properties of the non-competitive NMDA receptor antagonist memantine, in combination with a positive allosteric modulator of metabotropic glutamate receptors of Group III, VU 0422288. The treatment was started 48 h after the injection of neurotoxic agent trimethyltin (TMT) at 7.5 mg/kg. Three weeks after TMT injection, functional and morphological changes in a rat hippocampus were evaluated, including the expression level of genes characterizing glutamate transmission and neuroinflammation, animal behavior, and hippocampal cell morphology. Significant neuronal cell death occurred in the CA3 and CA4 regions, and to a lesser extent, in the CA1 and CA2 regions. The death of neurons in the CA1 field was significantly reduced in animals with a combined use of memantine and VU 0422288. In the hippocampus of these animals, the level of expression of genes characterizing glutamatergic synaptic transmission (Grin2b, Gria1, EAAT2) did not differ from the level in control animals, as well as the expression of genes characterizing neuroinflammation (IL1b, TGF beta 1, Aif1, and GFAP). However, the expression of genes characterizing neuroinflammation was markedly increased in the hippocampus of animals treated with memantine or VU 0422288 alone after TMT. The results of immunohistochemical studies confirmed a significant activation of microglia in the hippocampus three weeks after TMT injection. In contrast to the hilus, microglia in the CA1 region had an increase in rod-like cells. Moreover, in the CA1 field of the hippocampus of the animals of the MEM + VU group, the amount of such microglia was close to the control. Thus, the short-term modulation of glutamatergic synaptic transmission by memantine and subsequent activation of Group III mGluR significantly affected the dynamics of neurodegeneration in the hippocampus.

## 1. Introduction

Neurodegenerative conditions of different etiologies develop as a result of closely associated pathological processes, such as oxidative stress, mitochondrial dysfunction, and neuroinflammation. The main approaches to the treatment of neurodegenerative diseases, such as Alzheimer’s disease, Parkinson’s disease, amyotrophic lateral sclerosis, and so on, are based on developing an understanding of these mechanisms. Excitotoxicity is an important mechanism characteristic of most neurodegenerative diseases. It often leads to neuronal dysfunction and cell death due to the excessive extracellular release of glutamate and prolonged activation of its receptors [1]. Although glutamate does not directly kill neurons, elevated extracellular levels trigger cascades of reactions that damage neurons, leading to mitochondrial dysfunction, oxidative stress, and overproduction of reactive oxygen species [2]. It is generally accepted that ionotropic glutamate receptors, namely NMDARs, play a key role in this pathological mechanism due to their high permeability for Ca^2+^ ions [3,4]; therefore, the use of blockers of these receptors is considered one of the main ways to reduce excitotoxicity. Thus, the low-affinity, potential-dependent, non-competitive NMDA receptor antagonist memantine has been found to have a neuroprotective effect [5] and clinical applications [6].

Metabotropic glutamate receptors (mGluRs) have also attracted attention as targets for modulating the activity of the glutamatergic system. To date, eight subtypes of mGluRs have been characterized, which are divided into three groups based on their sequence homology, involved cell signaling pathways, and agonist selectivity [7]. Some mGluR ligands were shown to have neuroprotective effects in vivo [8,9,10,11].

Group III mGluRs in the hippocampus are found predominantly on neuron presynapses. They previously attracted our attention due to the fact that during experimental neurodegeneration induced by trimethyltin chloride (TMT), the level of mGluR4 gene expression in the hippocampus significantly increased 1, 3, and 6 weeks after TMT injection [12]. Given this, we decided to investigate the neuroprotective potential of positive allosteric modulators (PAMs) of Group III mGluRs (which include mGlu7 and mGlu8 as well as mGlu4) when administered together with the non-competitive NMDA receptor antagonist memantine. We believe that such modulation of the activity of glutamatergic synapses will contribute to the effective control of glutamate excitotoxicity and, as a consequence, to the survival of neurons.

We used the TMT model of neurodegeneration in our study. This neurotoxicant is considered a valid tool for the in vivo modeling of Alzheimer’s disease and similar neurodegenerative conditions [13]. Among the advantages of this model is the diversity of pathological processes induced by TMT in different areas of the brain at different time periods. Its effects include calcium overload, glutamate excitotoxicity, neuroinflammation, oxidative stress, and mitochondrial dysfunction [14,15]. The multiplicity of mechanisms of action of TMT on the brain makes it possible to study various pathological cellular processes (for example, excitotoxicity and neuroinflammation) in a complex manner, taking into account their interrelationships in vivo.

## 2. Results

Experiments were carried out on five groups of rats comprising the “Control” group and four treatment groups: TMT, TMT + MEM, TMT + VU, and TMT + MEM + VU (Figure 1). All treatment group animals received one intraperitoneal injection of 7.5 mg/kg TMT on day 0. The TMT + MEM group received memantine 48 and 72 h after TMT injection. The TMT + VU group received VU0422288 (a positive allosteric modulator of mGluR Group III) on days 5, 6, and 7. The TMT + MEM + VU group received both treatments at the same time points.

### 2.1. Nissl Staining

In the brain sections obtained from the control and experimental animals, a morphometry of the individual fields of the hippocampus was performed. Neurons in the control animals had an evenly stained cytoplasm and nucleoplasm, pronounced plasmatic and nuclear membranes, and clearly pronounced nucleolus. The cells were arranged closely and had a consistent size. The neurotoxicant TMT caused marked neuronal death. Common to all experimental groups was the most pronounced degradation of neurons of the pyramidal layers in the CA4 (Figure 2E) and CA3 (Figure 2D) regions of the hippocampus. A span of distorted tissue penetrated by glial cells (recognized by their smaller nuclei) was observed in the CA4 and CA3 regions. Pyramidal cells in those areas were either missing or often had dysmorphic shapes such as elongated triangular or rhomboid, with some ballooned cells with cytoplasmic achromasia. Such cells were not counted in the analysis, unless features of the nucleus could still be recognized.

Neurons of the CA2 field were the least vulnerable to the toxic effect of TMT. We found a few cells that were distorted in the same manner as in other regions, but there was no significant difference in the counts of cells per unit length of the region between the control and experimental groups of animals in this region (Figure 2C).

Of particular interest are the results obtained in the CA1 region of the hippocampus. Exposure to TMT led to the death of some neurons in the CA1 region. As a result, the layer of pyramidal neurons in this field in the experimental animals was visually thinner than in the control rats. Vacuoles and compacted detritus-like masses were often seen instead of healthy neurons.

From calculating the density of healthy neurons in the CA1 field, we found that for the animals in the TMT and TMT + VU groups, cell density was significantly lower than in the control animals. Similarly, cell density in TMT + MEM + VU was significantly different from both the TMT and TMT + VU groups, and was not different from the control. The TMT + MEM group in the CA1 region showed no difference from any of the other groups. While significance was not reached for the TMT + MEM group, its median values suggest intermediate efficiency in preserving neuron density in the CA1 region, compared to the TMT + MEM + VU group (Figure 2B). The result indicates that the combined administration of memantine and VU0422288 is capable of improving the preservation of neurons in the CA1 region of the hippocampus.

### 2.2. Passive Avoidance Test

Local damage to the hippocampus leads to behavioral disorders. The passive avoidance reflex is one of the basic ones used in assessing the influence of substances on the formation and reproduction of a memory trace. The death of neurons in the hippocampus affected the memory of the animals administered with TMT, affecting the formation and reproduction of memory. According to the results obtained, memory was impaired in all experimental animals (Figure 3). The test was concluded either once the animal had entered the dark chamber or 180 s after the beginning of the test.

### 2.3. qRT-PCR Analysis

From studying the expression levels of genes characterizing glutamatergic synaptic transmission in the hippocampus, we found that TMT and applied pharmacological interventions did not have a significant effect on them (Figure 4, upper panel). This mainly refers to the mRNA expression of EAAT2 (encoding the glial glutamate transporter) which is responsible for 90% of glutamate reuptake [16]. The levels of expression of the subunit of NMDA receptors Grin2B and the subunit of AMPA receptors Gria1 were also close to the control values, except for the level of expression in animals of the TMT + VU group, in which it was reduced.

In addition, we examined the expression of genes involved in neuroinflammation, Aif1, Il1b, GFAP, and TGFb1 (Figure 4, lower panel). The level of expression of the Aif1 gene encoding the IBA1 protein, a recognized marker of microglia, significantly increased only in the TMT + VU group relative to the control. The level of Il1b mRNA encoding one of the pro-inflammatory cytokines IL-1 beta was increased in the hippocampus of animals of all experimental groups compared to the control, except for the TMT + MEM + VU group, in which the level did not differ from the control. Expression of the GFAP gene encoding a protein of the same name and a recognized marker of astrocytes increased three or more times relative to the control in the TMT, TMT + VU, and TMT + MEM groups, but not in the TMT + MEM + VU group. The level of TGFb1 encoding transforming growth factor beta 1, one of the anti-inflammatory cytokines, was most elevated compared to the control in the TMT + VU subgroup, as well as in the TMT + MEM and TMT subgroups, but not in the animals of the TMT + MEM + VU group (Figure 4).

The obtained results show a pronounced neuroinflammation in the hippocampus of animals of all experimental groups, except for TMT + MEM + VU.

### 2.4. Immunohistochemistry

Observed changes in the gene expression of the neuroinflammation markers in the hippocampus gave rise to the question: in which regions of the hippocampus is the focus of inflammation located and how does the proposed pharmacotherapy affect it?

The results of the immunohistochemical staining showed that microglial cells in the animals of the control group were regularly distributed in the tissue and had a branched phenotype with compact cell bodies and elongated processes. Meanwhile, in the experimental groups, we observed an increase in the number of activated amoeboid microglial cells, featuring thickened or absent processes, mainly in the hilus and CA4–CA3 fields of the hippocampus, as well as increases in rod-shaped microglia in the CA1 pyramidal layer of the hippocampus (Figure 5A).

The results of counting IBA1-positive cells showed that the density of microglial cells in the hilus was significantly higher in the TMT and TMT + MEM groups compared to the control group (Figure 5B), while in the CA1 pyramidal layer, there was no significant difference between groups (Figure 5E). The ratio of the area occupied by IBA1-positive cells to the total area reflects the degree of microglial activation. According to this parameter, we found that in the hippocampal hilus in the TMT group, the degree of activation was significantly higher than in the control group (Figure 5C), while there were no significant differences between the groups in the pyramidal CA1 field (Figure 5F). Besides counting the cells, we were able to analyze the shape of the microglial cell bodies observed in the tissue. To describe the shape, we used the Feret aspect ratio, useful for assessing an elongated phenotype, which is characteristic of rod-shaped microglia. The results showed no differences in this parameter in the hilus, which mainly featured thick clusters of ameboid microglia (Figure 5D); however, in the CA1 region, there were more elongated cells relative to the control in the TMT, TMT + MEM, and TMT + VU groups, but not in the TMT + MEM + VU group (Figure 5G).

The analysis of the GFAP-positive cell bodies in terms of their cell density and area occupied did not reveal significant differences between any of the groups in the hilus or in the CA1 region. The overall presence of GFAP-positive labeling, evaluated using automated thresholding with Otsu’s algorithm, which should have accounted for cell processes as well as cell bodies, did not show significant differences either.

## 3. Discussion

Targets of pharmacological action aimed at reducing neurodegeneration include glutamatergic synaptic transmission, the modulation of which is important for the control of excitotoxicity, which is relevant for the survival of neurons. As shown by the results of this work, under conditions of neurodegeneration that had already begun, short-term inhibition of NMDA receptors and a decrease in the release of glutamate into the synaptic cleft actually reduced the neurotoxic death of neurons in the hippocampus.

Experimental models of neurodegeneration caused by the neurotoxicant TMT have been studied quite well, and many pathophysiological processes in the brains of animals and humans have been described at different stages of action of this neurotoxicant. Unlike other, more indiscriminate chemical agents, TMT primarily damages the limbic system, particularly the hippocampus, while unobstructed by an intact blood–brain barrier [17]. The hippocampus is closely associated with attention, memory acquisition, learning processes, and spatial orientation [18,19,20]. Among the reasons for the high vulnerability of the hippocampus to adverse effects of TMT is the massive damage to the mitochondria in this brain structure [21]. Damaged mitochondria trigger microglial activation as well as autophagy, most pronounced in the hippocampus one week after TMT intoxication. These cascades remain active in the hippocampus at later time points as well, even after the one-time administration of TMT [12,21]. In the present study, we assessed the morphological and functional state of the hippocampus 3 weeks after TMT injection. By this time, some of the pathological processes in the hippocampus had subsided, for example, mitochondrial dysfunction [21], as well as excitotoxicity [22], which was confirmed by the results of this work (Figure 4, top panel). The dominant pathological process at that time point appeared to be neuroinflammation (Figure 4, bottom panel, and Figure 5).

It has been previously shown, both in our laboratory and by other researchers, that the behavioral consequences of TMT were hyperactivity in an open-field test, increased locomotor activity, and deficits in passive and active avoidance behavior, as well as changes in T-maze and Morris water maze behavior [12,23]. Several neuroprotective strategies have been successfully shown in the TMT model, as highlighted in reviews [13,17,24].

In this work, using an in vivo neurodegeneration study for the first time, we show the joint short-term modulation of the activity of glutamatergic synapses by an NMDAr antagonist and mGlu glutamate receptor PAM. The results show that such treatment can reduce the death of neurons in the hippocampus. As expected, 3 weeks after the injection of TMT, pronounced death occurred in individual fields of the hippocampus in varying degrees: hilus > CA3 > CA1 > CA2.

The work by Dragic [25] showed a similar distribution of the degree of neuronal death under the influence of TMT at a dose of 8 mg/kg. Assuming the possibility of improving the survival of neurons in the CA1 region, we assessed the survival of neurons in this field of the hippocampus. Indeed, this evaluation revealed that the combined use of memantine and VU0422288 had a significant neuroprotective effect (Figure 2B). However, the loss of neurons in the hilus and CA3 regions still inevitably leads to the disruption of hippocampal function, which was evident in the behavioral experiments. It is known that the CA3 field of the hippocampus is required for some forms of learning, in particular for rapid contextual learning in one attempt [26]. The disturbance of passive avoidance in animals due to the action of TMT, shown in our work, confirms this point of view. It is interesting that the animals from two groups introduced to VU0422288 showed some improvement during the passive avoidance test, though insignificant.

This dose of memantine was chosen based on the literature data showing that 5–20 mg/kg i.p. is the therapeutically relevant dose in rat experiments [27]. Memantine at these doses attenuated seizure activity [28] and L-DOPA-induced dyskinesia [29] and protected against Aβ-mediated memory impairment [30]. The neuroprotective effect of memantine, for example, has been shown in neuroinflammation experimentally induced by the intraventricular infusion of LPS in rats [31]. Memantine has also demonstrated a protective effect in cell cultures against 6-hydroxydopamine (OHDA)-induced damage [32]. However, under our conditions, memantine did not show noticeable neuroprotective properties and did not reduce TMT-induced neuronal death in the hippocampal CA1 field (Figure 2B). Note that we only administered two injections of memantine 48 and 72 h after TMT, and it is evident that this amount of treatment was not enough to significantly affect glutamatergic synaptic transmission. However, additional activation of Group III mGluR resulted in neuroprotection, and in animals of the MEM + VU0422288 group, the density of neurons in the hippocampal CA1 field did not differ from that in control rats (Figure 2B).

Group III mGluR ligands have many properties that have attracted attention as targets for neurological disease therapy [9,33]. For example, the mGlu4 positive allosteric modulator, PHCCC, and the mGlu7 allosteric agonist AMN082 provided relief from akinesia in a reserpine-treated rat model of Parkinson’s disease, which the authors believed most likely reflected the inhibition of excess glutamate release [34].

The role of mGlu receptors is not limited to their involvement in the mechanisms of glutamate excitotoxicity; they are also involved in other processes observed during neurodegeneration, such as neuroinflammation and oxidative stress [9,35]. Activation of Group III mGlu receptors inhibited the inflammatory process in cultured glial cells [36]. The substance VU0422288 is a PAM of the mGlu III receptor group, first described in 2014 [37]. To date, this substance has not been previously described in the context of a possible treatment for neuroinflammation. Existing works that mention the effects of VU0422288 concern synaptic transmission and the potential of a PAM in the treatment of depression, autism, ADHD, and schizophrenia [37], as well as its potential for the treatment of Rett syndrome [38]. In our work, TMT caused large increases (>3-fold) in the hippocampal levels of GFAP mRNA, a marker for astrocytes (Figure 4, bottom panel). An increase in the expression levels of this protein indicates active astrogliosis in the hippocampus three weeks after TMT injection. In contrast to the other experimental groups, in the TMT + MEM + VU group of animals, the level of GFAP expression did not differ from the control group level.

The effects of the pharmacological substances we applied significantly affected the course of neuroinflammation. This is evidenced by the levels of expression of markers AIF1, IL1b, and TGFb1, which did not differ from the levels of the control animals (Figure 4, bottom panel). The results of the immunohistochemical study of microglia also indicated a decrease in neuroinflammation in the hippocampus of rats in the TMT + MEM + VU group.

It is known that microglia are activated in rats on the first day after intoxication, and this activation persists for a long period [39]. The results of this work show a significant activation of microglia three weeks after TMT. Activated microglia had a different shape from that of the control. Usually, it is described as rod-shaped and amoeboid, depending on the stage and location of activation. In contrast to the hilus, in the CA1 field, microglia were found to be predominantly rod-shaped. Rod-shaped microglia are commonly found in the early stages of neurodegenerative disorders and trauma [40]. Therefore, it can be assumed that neurodegeneration in the CA1 field would have continued in the future, and the pharmacological actions taken would only slow down the neurodegenerative process in the hippocampus. For its decisive suppression, the use of anti-inflammatory substances would likely have been necessary.

The results obtained in this work confirm the hypothesis that the control of glutamate excitotoxicity affects the course of neuroinflammation. Under our experimental conditions, short-term modulation of glutamatergic synaptic transmission had an effect on neuroinflammation, although its complete resolution did not occur. A decrease in cell death from the combined use of an NMDA receptor antagonist and PAM of mGluR Group III was observed, as compared to the separate use of these substances. This result seems to be due to the fact that the substances target both pre- and postsynaptic segments of glutamate neurotransmission. Currently, the clinical use of glutamate metabotropic receptor ligands is limited [41]. Nevertheless, we believe that the fine modulation of glutamatergic synaptic transmission using mGluR ligands remains promising. Our work shows the prospect of a joint action on the pre- and postsynaptic sides of glutamatergic neurotransmission in order to modulate pathophysiological processes.

### Limitations

The results of this study cannot be easily applied to clinical work and only offer initial values for translational research in the treatment of neurodegenerative diseases. We did not investigate the effects of memantine and VU0422288 at different concentrations or during longer regimens. Nevertheless, our study provides the necessary information for further experiments with these drugs.

Among the questions that remain to be answered in the future is whether the reduction in cell death we identified in the hippocampus is sustainable and will continue in the long term. Did neurodegeneration stop in the CA1 field of the hippocampus or did it only slow down? Lastly, would the effects we found be valid for neuroprotection when using other experimental models of neurodegeneration, as well as in other areas of the brain (for example, in the neocortex)?

## 4. Materials and Methods

### 4.1. Materials

The following drugs and chemicals were provided by or obtained from the sources indicated: trimethyltin (TMT, CAS #1066-45-1; Sigma-Aldrich Co., St. Louis, MO, USA); memantine hydrochloride (MEM, Cat. No. #0773; Tocris, Bristol, UK); PAM Group III mGluRs N-[3-chloro-4-[(5-chloro-2-pyridinyl)oxy]phenyl]-2-pyridinecarboxamide (VU0422288, Cat. No. #05378; Tocris, Bristol, UK).

### 4.2. Animals and Treatments

In the experiments, male Wistar rats were used, whose weight at the beginning of the experiments was 210 ± 10 g. The animals were kept and used in accordance with the rules of the Council of the European Community (directive of 1986) and the protocol was approved by the Commission on Biological Safety and Ethics at the Institute of Theoretical and Experimental Biophysics, Russian Academy of Sciences (February 2021, Protocol No. 7/2021). All experiments were performed according to the procedures approved by the local animal control authorities. The animals were kept under standard conditions, including a 12 h:12 h light/dark cycle. Food and water were available ad libitum.

All animals (*n* = 35) were arbitrarily assigned to 5 groups of 7 by the experimenter and identified by a number (Figure 1); the experimenter was not blind to the groups. For the detailed study design, see the flowchart (Figure 1). TMT was administered to rats of all “treatment” groups in their home cages as follows: TMT (7.5 mg/kg, i.p., single injection) (*n* = 28) and physiological saline was administered to the “control” group of animals (*n* = 7) on day 0. Pharmacological suppression of excitotoxicity was carried out using 2 injections of the non-competitive NMDA glutamate receptor antagonist memantine hydrochloride at a dose of 10 mg/kg on the second and third days after TMT (*n* = 7) or by 3 injections of a positive allosteric modulator of metabotropic glutamate receptors Group III VU 0422288 at a dose of 3 mg/kg on the fifth, sixth, and seventh days after TMT injection (*n* = 7), or combined pharmacological suppression of excitotoxicity was carried out using 2 injections of memantine hydrochloride at a dose of 10 mg/kg on the second and third days after TMT and 3 injections of a VU 0422288 at a dose of 3 mg/kg on the fifth, sixth, and seventh days after TMT injection (*n* = 7). Animals were sacrificed by decapitation (n_animals_ = 20, gene expression analysis, and Nissl staining) or transcardiac perfusion (n_animals_ = 15) with saline followed by 4% paraformaldehyde (immunohistochemistry) beginning with the treated groups and followed by the saline controls. Animals undergoing PFA perfusion were terminally anesthetized with a mixture of xylazine and zoletil (tiletamine-zolazepam) at a dose of 50 mg/kg prior to the procedure. No additional medications or analgesics were given to the animals pre- or postdosing to reduce pain and/or suffering as this study only involved neuroactive compounds at doses that do not elicit pain.

### 4.3. Passive Avoidance Test

To assess passive avoidance in the animals of all groups, tests were carried out using the hardware–software complex “Shelter” (version 2.6.1; Neurobotics, Moscow, Russia). On the 10th through 12th days after TMT administration, each animal was placed in the light compartment of the device, where they were given the opportunity to examine their surroundings for 180 s. During the training, each animal was placed in the light compartment, and after they moved to the dark part, an electric current of 2 mA was applied to the floor grid for 2 s. After that, the animals were returned to their home cages. The retention of memory was tested 7 days after training (21 days after TMT). The latency to enter the dark compartment was recorded.

### 4.4. Specimen Harvesting

After completion of behavioral tests, three of the seven animals in each subgroup were anesthetized with a mixture of xylazine and zoletil (tiletamine-zolazepam) at a dose of 50 mg/kg and transcardially perfused by a cold phosphate-buffered saline (PBS) followed by 4% paraformaldehyde (PFA). Brains were removed and postfixed in paraformaldehyde for 24 h and later used for immunohistochemistry (IHC). The remaining four out of seven animals from each group were decapitated, and the brain was isolated from each rat and bisected into its hemispheres. One of the hemispheres was fixed in Carnoy’s solution for 5 h, followed by dehydration by subsequent incubations in alcohols of increasing concentrations, chloroform, and then impregnated with paraffin and embedded in paraffin blocks. The hippocampus was isolated from the second hemisphere of the brain, placed in a denaturing buffer (4 M guanidine thiocyanate, 25 mM Na_3_O_7_C_6_H_5_, 0.5% N-laurylsarcosine sodium salt, 0.1 M 2-mercaptoethanol), homogenized, and stored at −20 °C to be used for assessing the expression levels of target genes by qRT-PCR analysis.

### 4.5. Histological Examination

#### 4.5.1. Nissl Staining

The paraffin blocks were cut serially into thin sections (8 µm) and the sections were further stained with Nissl-modified toluidine blue (BioVitrum #07-003) according to the manufacturer’s instructions. Bright-field microscopy was carried out using an Olympus (IX71; Olympus Corporation, Tokyo, Japan) microscope. A 40× dry objective was used to collect bright-field images, digitized using a Canon camera mounted on a designated chassis port. To create a panoramic image of the brain slice, a series of successive images was assembled using ImageJ 1.53f software, which was also used for analysis. Panoramic images of the hippocampal pyramidal layer were straightened into a line using the Straighten package of Fiji/ImageJ. The resulting images were segmented for intact cells using CellPose 2.0 and manual verification. The metrics from both methods were extracted using Fiji ImageJ.

#### 4.5.2. Immunohistochemistry

Floating, 50 µm thick fixed brain sections were prepared using a vibratome (VIBRATOME 100; IMEB). Sections were immunohistochemically stained by primary antibodies to the microglial marker IBA-1 (Rb #178846; Abcam) and astrocyte marker GFAP (Ch #4674; Abcam) at ratios of 1:2000 and 1:500, respectively. Secondary antibodies conjugated with Alexa 488 (Gt x Rb #150077; Abcam) and Alexa 647 (Gt x Ch #150171; Abcam) were used at ratios of 1:500 each. The samples were placed under coverslips in a ProLong Glass Antifade Mountant (Invitrogen, Waltham, MA, USA). IHC microscopy was performed using the Leica Thunder DMI8 fluorescence microscopy system (Leica Microsystems GmbH, Wetzlar, Germany) using a 40× dry objective (HC PL APO 40x/0.95 DRY). Bit depth was 12 bits. The data were collected as a set of 1024 × 1024 tiles arranged in a grid over the hippocampus, with 11% overlap at the margins, with a calculated thickness of the imaging plane of 4 µm. Leica’s proprietary ICC postprocessing method was used to reduce the signal from out-of-focus imaging planes. The tiles were stitched in a panoramic image using the Leica LAS X Mosaic Merge tool. Subsequent postprocessing of images was performed using the Fiji ImageJ v.1.53q package [42]. A section of hippocampus of a known area was selected in the CA1 and hilus regions. Cell body segmentation was conducted using CellPose 2.0 [43], utilizing a cell shape recognition model separately trained for each antibody used. Each model accounted for cell type-specific antibody signals as well as Hoechst staining of the cell nucleus. The results of the automatic segmentation were manually curated. The astrocyte area was evaluated using the automatic thresholding method described in [44] as implemented in ImageJ.

### 4.6. qRT-PCR Analysis

Total RNA was isolated from homogenates using a single-step method of extraction with an acid guanidinium thiocyanate-phenol-chloroform mixture [45]. To remove residual DNA, the samples were treated with DNase I (New England Biolabs, Ipswich, MA, USA) according to the manufacturer’s instructions. RNA quality was estimated electrophoretically in 1% agarose gel. The concentration of purified RNA was determined using an ND-1000 spectrophotometer (NanoDrop Technologies Inc., Wilmington, DE, USA). One microgram of the total RNA was used per reverse transcription reaction by Thermo Scientific RevertAid Reverse Transcriptase (#EP0442; Thermo Fisher Scientific, Vilnius, Lithuania). Samples with RNA subjected to reverse transcription without the reverse transcriptase were used as the negative controls. Synthesized cDNA was used for PCR with gene-specific primers (Table 1). Real-time PCR was carried out using the DNA amplifier DTlite (DNA-Technology, Moscow, Russia) with a qPCR Mix HS SYBR Kit (Evrogen, Moscow, Russia), from which SYBR Green I was used as an intercalating dye. The PCR program included 5 min of initial denaturation at 95 °C, followed by 35 cycles: 95 °C for 1 min, 58 °C for 30 s, and synthesis at 72 °C for 45 s. The fluorescence of SYBR Green I was measured for 15 s at the end of each cycle. The threshold cycle (Ct) value was determined using DTMaster software (DNA-Technology, Russia). There were 4 rats in each group, with three samples obtained from each animal. The signals were normalized to those obtained for the gene of the cytoskeletal protein beta-actin (Actb). Amplicon quality and sizes were verified by gel electrophoresis in 3% agarose gel. Relative quantification of gene expression was performed using the 2^−ΔΔCt^ method [46]. Changes in mRNA expression levels were calculated after normalization to Actb. The ratios obtained after normalization are expressed as the fold change over the corresponding saline-treated controls.

### 4.7. Statistical Analysis

Data are presented as the mean ± SEM. Statistical tests and graphs were created using GraphPad Prism software or RStudio v1.4.1106. Sample distributions were tested for normality using the Shapiro–Wilk test. Significance was tested using one-way ANOVA followed by Tukey’s multiple comparisons test or Kruskal–Wallis testing followed by Dunn’s multiple comparisons test in the case of a non-normal distribution (indicated in the text or in the captions of the figures).

## 5. Conclusions

The neurotoxicant TMT induces neurodegeneration in the hippocampus, causing learning and memory impairment in rats. Three weeks after TMT administration, the most noticeable neuronal death was observed in the CA4 and CA3 regions, accompanied by microglial activation. Less pronounced cell death occurred in the CA1 region, but was insignificant in the CA2 field. Co-administration of the non-competitive NMDA receptor antagonist memantine and Group III PAM mGluR VU0422288 reduced neuronal death in the CA1 region and also reduced TMT-induced neuroinflammation.

## Figures and Tables

**Figure 1 ijms-24-08249-f001:**
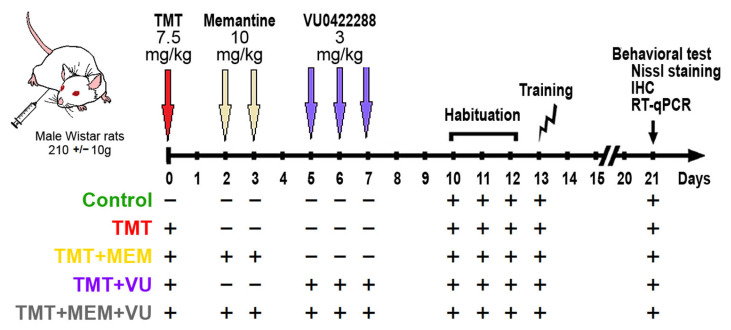
Experimental study design. All animals were arbitrarily assigned to 5 groups: “Control”—sham injection (0.9% NaCl), green color (*n* = 7); “TMT”—trimethyltin chloride (7.5 mg/kg, once), red color (*n* = 7); “TMT + MEM”—TMT (7.5 mg/kg, once) + memantine hydrochloride (10 mg/kg, twice), yellow color (*n* = 7); “TMT + VU”—TMT (7.5 mg/kg, once) + VU 0422288 (3 mg/kg, three times), violet color (*n* = 7); and “TMT + ME + VU”—TMT (7.5 mg/kg, once) + memantine hydrochloride (10 mg/kg, twice) + VU0422288 (3 mg/kg, three times), gray color (*n* = 7). After behavioral tests, the brains of animals of all groups were isolated 21 days after TMT injection for further cellular and molecular studies by Nissl staining, immunohistochemistry, and RT-qPCR.

**Figure 2 ijms-24-08249-f002:**
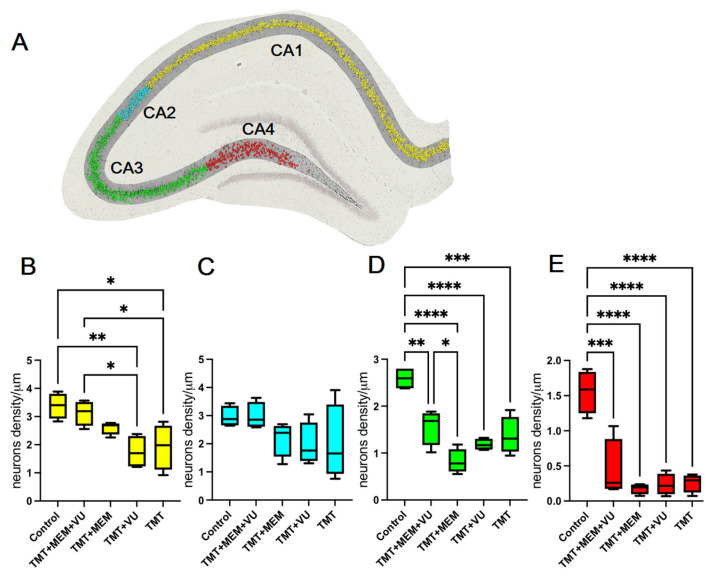
Nissl staining results. (**A**)—A slice of the hippocampus from the control group with colors highlighting segmented pyramidal neurons in each region of the hippocampus; red—CA4, green—CA3, cyan—CA2, and yellow—CA1. (**B**–**E**)—Neuron counts, represented as cell density per unit length. Statistical significance was assessed using one-way ANOVA followed by Tukey’s multiple comparisons test, * *p* < 0.05, ** *p* < 0.01, *** *p* < 0.001, and **** *p* < 0.0001. *n*(rats) = 4 in each group; *n*(cells) = CA1–11941, CA2–1508, CA3–3731, CA4–1046. Effect sizes (η^2^) for each test, respectively = CA1: 0.66, CA2: 0.31, CA3: 0.84, CA4: 0.85.

**Figure 3 ijms-24-08249-f003:**
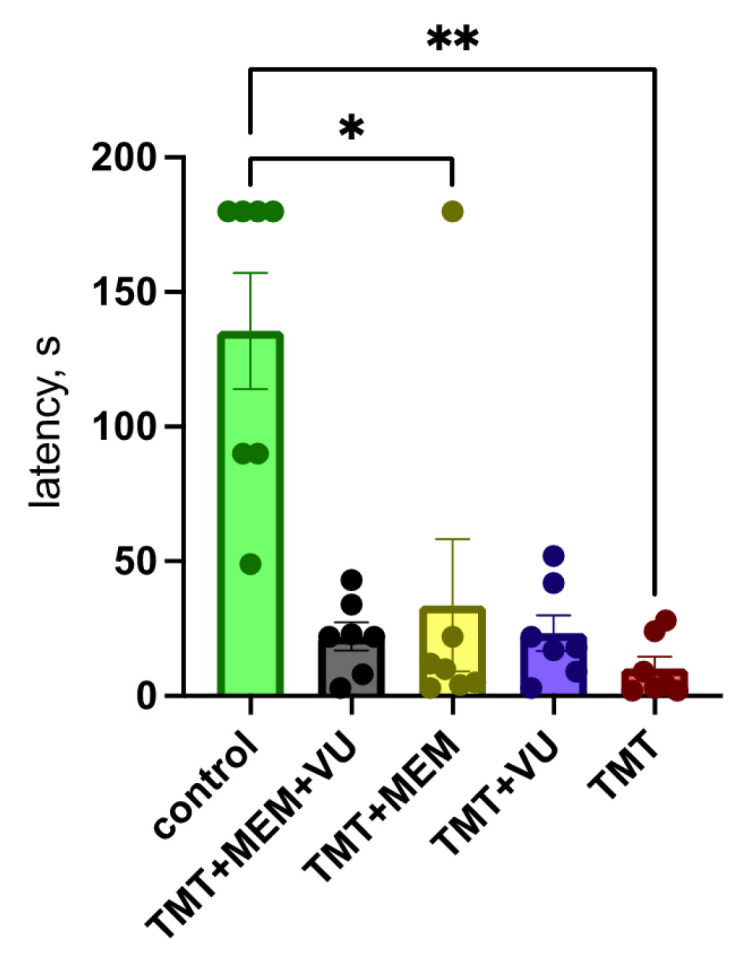
Step-through passive avoidance test, latency to enter the dark compartment, 7 days after training. TMT-treated group (*n* = 7), “TMT + MEM” group (*n* = 7), “TMT + VU” group (*n* = 7), “TMT + MEM + VU” group (*n* = 7), and “Control” group (*n* = 7). Data are presented as the mean ± SEM. (Kruskal–Wallis test; *p* = 0.0022. Dunn’s multiple comparisons test; *—*p* = 0.0193; **—*p* = 0.0013).

**Figure 4 ijms-24-08249-f004:**
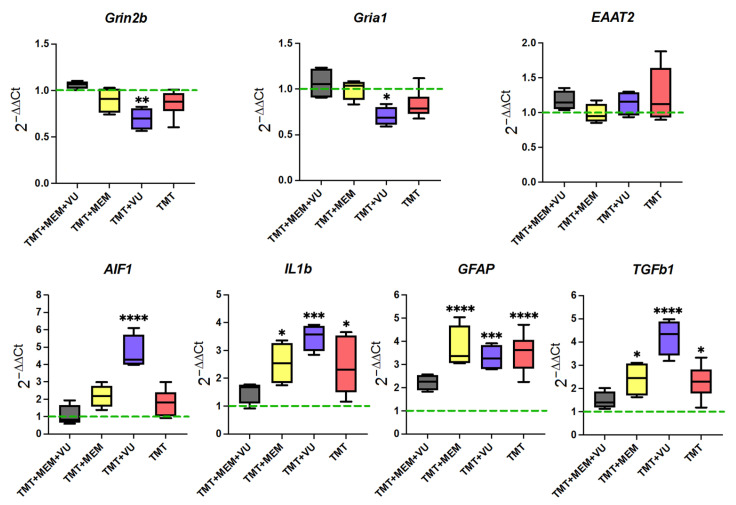
Effects on the mRNA expression of proteins involved in glutamate systems (**top** panel) and inflammation (**bottom** panel) in the TMT and treatment groups. The control group expression level is indicated by the dashed green line. Statistical significance was assessed using one-way ANOVA followed by Dunnett’s multiple comparisons test, * *p* < 0.05, ** *p* < 0.01, *** *p* < 0.001, and **** *p* < 0.0001.

**Figure 5 ijms-24-08249-f005:**
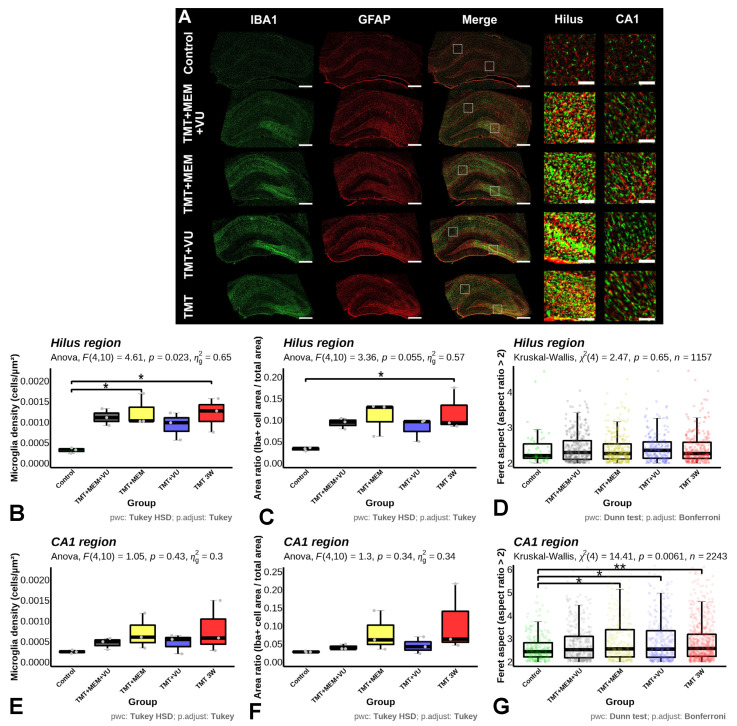
Effects of TMT and subsequent treatment with MEM or VU, or a combination of MEM + VU, on microglial and astrocyte morphology, 21 days after TMT injection. (**A**)—Representative micrographs of hippocampal brain sections showing localization and morphology of microglia and astrocytes. A green channel was used to identify microglia (IBA1-positive cells) and red, to identify astrocytes (GFAP-positive cells). Scale bars—500 µm for mosaic images of whole slices of hippocampus and 100 µm for fragments of DG or CA1 regions; (**B**,**E**)—microglial density in the hilus and CA1 regions of the hippocampus; (**C**,**F**)—area ratio of IBA1-positive cells in the hilus and CA1 regions of the hippocampus; (**D**,**G**)—Feret aspect ratio >2 IBA1-positive cells in the hilus and CA1 regions of hippocampus. The “area ratio” is defined as the ratio between the sum of the areas of the cell body projections to the total area of the region of interest, representing the “packing” of the region. The Feret aspect ratio is the ratio between the longest Feret caliper to the shortest Feret caliper of each cell. Cells with a Feret aspect ratio below 2 were discarded from analysis to emphasize the presence of rod-like microglia. Statistical tests are indicated in figure captions, * *p* < 0.05, ** *p* < 0.01.

**Table 1 ijms-24-08249-t001:** Oligonucleotides used for qRT-PCR. The design of the oligonucleotides was carried out in Primer-BLAST (www.ncbi.nlm.nih.gov, accessed on 5 August 2021).

*Gene* (Protein)	Oligonucleotide 5′–3′ (Forward + Reverse)	Amplicon Size (bp)
*Actb* (beta-actin)	ATGGTGGGTATGGGTCAGAACTTTTCACGGTTGGCCTTAG	225
*Grin2b* (receptor subunit GluN2B of the NMDA)	AATGGCGGATAAGGATGAGTC CTTAGAGTCGCCATCGTCCA	247
*EAAT2* (glutamate transporter-1)	GAGGAGGCCAATACAACCAA TTCATCCCGTCCTTGAACTC	305
*Aif1* (allograft inflammatory factor 1)	TCATCGTCATCTCCCCACCTA AACTCCATGTACTTCGTCTTGAA	117
*Gfap* (glial fibrillary acidic protein)	CGAAGAAAACCGCATCACCA CCGCATCTCCACCGTCTTTA	239
*IL1b* (interleukin-1 beta)	GAAGAAGAGCCCGTCCTCTGTG ATGGGTCAGACAGCACGAGGC	127
*TGFb1* (transforming growth factor beta 1)	AGAGCCCTGGATACCAACTA GACCTTGCTGTACTGTGTGT	186

## Data Availability

The data presented in this study are contained within this article.

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
