# Peer review of "Pharmacological Modulation of Excitotoxicity through the Combined Use of NMDA Receptor Inhibition and Group III mGlu Activation Reduces TMT-Induced Neurodegeneration in the Rat Hippocampus"

_ijms, 2023, doi:10.3390/ijms24098249_

Round 1
Reviewer 1 Report
Authors have provided the solid study concenrning the neuroprotective effects of mGluIII positive modulation in pathologies causes by trimethylin. This modulation strongly antagonises proapoptotic effect if glutamate activation of NMDA, mGlu1 and mGlu5 receptors. The combination of animal behavior and molecular data say for a possible potential of mGluIII allosteric modulators for medical treatment.
I have only minor comments concerning the study:
1) Line 80. It is the first mention of VU 0422288 in the text. It will be good to mention, that this substance is group III mGlu receptor positive allosteric modulator.
2) Line 106. It is not clear what kind of cell distotion occurs.
3) Line 239. Please provide the reference on the work describing more prominent changes in hippocampus as compared to other brain regions.
Author Response
Response to Reviewer #1:
I have only minor comments concerning the study:
- Line 80. It is the first mention of VU 0422288in the text. It will be good to mention, that this substance is group III mGlu receptor positive allosteric modulator.
Thank you for pointing this out for clarity, done.
- Line 106. It is not clear what kind of cell distortion occurs.
The changes are similar to other regions, but affecting less cells. Updated in text.
- Line 239. Please provide the reference on the work describing more prominent changes in hippocampus as compared to other brain regions.
Thank you, updated in text. Unlike other chemical agents, TMT primarily damages the limbic system, particularly the hippocampus, while unobstructed by an intact blood–brain barrier [Lee et. al, Trimethyltin-Induced Hippocampal Neurodegeneration: A Mechanism-Based Review, 2016]
Reviewer 2 Report
The manuscript is designed and written well. However, the Figures are not enough readable, please, use other format of figures (for example Figure 5, lower part).
Author Response
The manuscript is designed and written well. However, the Figures are not enough readable, please, use other format of figures (for example Figure 5, lower part).
Thank you for the note; we've redesigned the lower part of Figure 5 to be more readable.
Reviewer 3 Report
Neurodegeneration involves multiple faculties and hence requires a multitargeted approach. The current manuscript explores the possibility of using an NMDA antagonist and PAM, VU0422288, a Group III mGluR agonist in vivo. To achieve this, the authors inject male Wistar rats with TMT to mimic the neurodegeneration in the hippocampus. They inject some mice with memantine/ VU 0422288/ both, and compare the treatment groups with the control in behavioral and immunohistochemistry experiments. Using neuronal density measurements, they report a change in the CA1, CA3, and CA4 regions. They report no changes in memory impairment in comparison to TMT group. They also report a reduction in the NMDA and AMPA related gene Grin2b and Gria1, and an increase in inflammatory markers in the combined treatment groups. They also report an increase in microglial activation in the hilus, in the TMT+MEM group. They conclude that the combined treatment increases the effectiveness, and that the control of excitotoxicity effects neuroinflammation.
Overall,
1) The authors interpretations and conclusions are supported by the experiments.
2) Would using an AMPA antagonist in combination yield better outcomes?
3) How does the IL10 change with the treatment?
4) Are the differential effects seen in the CA1 through 4, due to varied expression of the mGluR/ Group III receptors?
5) The authors could include another research “The effects of predator odor (TMT) exposure and mGlu3 NAM pretreatment on behavioral and NMDA receptor adaptations in the brain” and include a few lines in the discussion.
6) Is there a reason the authors chose male rats?
Author Response
Response to Reviewer #3:
1) The authors’ interpretations and conclusions are supported by the experiments.
Thank you.
2) Would using an AMPA antagonist in combination yield better outcomes?
This is a valuable suggestion which requires additional consideration. On one hand, the blocking of AMPA receptors should decrease the excitotoxic presence of glutamate, therefore application of blockers should be synergetic with our treatment. On the other hand, data from the literature suggests a small neuroprotective potential of AMPAr antagonists, and possibility of increased side-effects, such as reduction of LTP and learning, or associated cognitive and emotional deficits.
3) How does the IL10 change with the treatment?
Previously, we have shown that, 21 days after TMT injection, the level of IL-10 mRNA in the hippocampus and prefrontal cortex was increased. However, 4 weeks after TMT injection, the content of anti-inflammatory IL-10 was increased in the prefrontal cortex of the rat brain, but not in the hippocampus. [Kamaltdinova, E., Pershina, E., Mikheeva, I. et al. Different Activation of IL-10 in the Hippocampus and Prefrontal Cortex During Neurodegeneration Caused by Trimethyltin Chloride. J Mol Neurosci 71, 613–617 (2021). https://doi.org/10.1007/s12031-020-01682-w]. It is known that IL-10 is a potent anti-inflammatory cytokine, which assists towards resolution of neuroinflammation.
4) Are the differential effects seen in the CA1 through 4, due to varied expression of the mGluR/ Group III receptors
In our opinion, the TMT-induced damage to hippocampus is not directly related to distributions of receptors among different hippocampal regions, because the primary target of TMT is believed to be the mitochondria, specifically stannin.
At the same time, the distribution of mGluR/III is indeed not equal across hippocampus regions. mGluR/III include mGlu4 and mGlu6-8 subtypes. From the literature, [Ferraguti, F., Shigemoto, R. Metabotropic glutamate receptors. Cell Tissue Res 326, 483–504 (2006). https://doi.org/10.1007/s00441-006-0266-5], the hippocampus appears to express mGlu4, 7 and 8 in different regions and at different levels. mGlu4 labeling is prominent in the CA1-3, in str. lacunosum moleculare and str. oriens. mGlu7a is seen through all dendritic layers, whereas mGlu7b was observed only in the terminal zone of the mossy fibres. The expression pattern of mGlu8 is more restricted than that of mGlu7. mGlu8 is marked in the terminal zones of the lateral perforant path, i.e. the outer layer of the CA3 str. lacunosum moleculare and the outer one third of the molecular layer of the dentate gyrus.
mGlu7a subtype is probably of the greatest interest in the context, because it dominates over other subtypes of group III receptors in terms of presence, and at the same time it is enriched in the CA1 zone. Considering this, we may assume the importance of mGlu7a receptor subtype contribution to the therapy. Selective targeting of this receptor subtype is a possible approach to improvement of neuroprotection.
5) The authors could include another research “The effects of predator odor (TMT) exposure and mGlu3 NAM pretreatment on behavioral and NMDA receptor adaptations in the brain” and include a few lines in the discussion.
The work suggested explores negative pharmacological modulation of the mGlu3 receptor subtype, which belongs to the mGluR group II (this group includes mGlu2 and 3 subtypes). In our work, we used a specific group III mGluR modulator, which includes mGlu4, 6, 7, and 8 subtypes and does not interact with group II mGluRs. The substance used in the suggested article also abbreviated as TMT (trimethylthiazoline), but is not related to trimethyltin chloride (also commonly abbreviated TMT). The work is also related to study of anxiety, but seems not to be related to neurodegeneration.
6) Is there a reason the authors chose male rats?
In our study, we used mature male Wistar rats to avoid the possible interference caused by the cyclical changes in the hormonal background characteristic for female animals.
Reviewer 4 Report
1. The reason for focusing on trimethyltin is not clear.
2. The research methods and results are considered excellent.
3. This research is expected to contribute to the treatment of Parkinson's disease in the future.
4. Materials and methods should be placed before the results.
Author Response
Response to Reviewer #4:
- The reason for focusing on trimethyltin is not clear.
The ability of TMT to induce neuronal death in certain areas of the brain has attracted attention for the creation of an experimental model that allows studying the cellular and molecular mechanisms of neurodegeneration, outlining approaches for creating new neuroprotective agents [Corvino, V., Marchese, E., Michetti, F. et al. Neuroprotective Strategies in Hippocampal Neurodegeneration Induced by the Neurotoxicant Trimethyltin. Neurochem Res 38, 240–253 (2013). https://doi.org/10.1007/s11064-012-0932-9]. Several advantages of using TMT as a tool to investigate the mechanisms of neurodegeneration are pointed out:
1) areas of the brain subject to damage, cellular targets and dynamics of neurodegeneration was well-described;
2) the blood-brain barrier does not undergo significant changes, which limits the contribution of peripheral factors to the mechanisms of neurotoxicity;
3) known protein and mRNA markers of neurodegeneration can be used as a control in the study of gene expression associated with the action of TMT;
4) neurodegeneration after a single injection of a neurotoxicant occurs in a chronic form, which makes it possible to investigate individual pathological phenomena characteristic for chronic disease;
5) TMT-induced neurodegeneration shares pathophysiological characteristics with several neurodegenerative diseases such as temporal lobe epilepsy and Alzheimer's disease.
- The research methods and results are considered excellent.
Thank you.
- This research is expected to contribute to the treatment of Parkinson's disease in the future.
Thank you.
- Materials and methods should be placed before the results.
The placement is related to requirements of a journal.
Reviewer 5 Report
The authors have dealt with a very interesting and often challenging subject that has translational significance. The pathophysiology of neurodegeneration and tackle treatments regarding Alzheimer’s disease.
Nevertheless, there are some points that need the immediate attention of the authors.
The is no translational or clinical hypothesis in the abstract regarding the results ie “The death of neurons in the CA1 field was significantly reduced in animals with combined use of memantine and VU 0422288.”, “However, the expression of genes characterizing neuroinflammation was markedly increased in the hippocampus of animals treated with memantine or VU 0422288 alone after TMT.”
Line 60-61, “PAM of group III mGluRs”: Please, analyze and explain about the substance and its role. It is the first time mentioned here.
Lines 125-128, “At the same time, the density of neurons in animals of the TMT+MEM and TMT+MEM+VU groups did not differ from that in the control group (Fig. 2B). The result indicates that the combined administration of memantine and VU0422288 is capable of improving the preservation of neurons in CA1 region of the hippocampus.”: Be very careful when jumping into results. In the figure there is difference between TMT and TMT+MEM+VU but not between TMT and TMT+MEM. Please rephrase the whole paragraph.
Lines 135-136, “According to the results obtained, memory was impaired in all experimental animals, to the greatest extent – in the TMT group (Fig. 3).” Be very careful of the phraseology. The lowest p-value does not mean greatest extent. Means lowest probability of making a statistical error… Please rephrase.
Method Section Lines 352-364: There is no power analysis regarding the number of the test subjects. There is also no reference regarding the dosage schema of the Memantine and the VU0422288. Why did not use injections through the whole experiment as it resembles more the clinical state (Patients with AD take Memantine for long periods of time).
Line 365 “animals were killed”: Replace with sacrifice. And also write the exact number of animals that were sacrificed by each method (decapitation and transcardiac perfusion) also in that part (3 PFA and 4 decapitation).
Lines 377-379 “On 11th through 13th days after TMT….”: In figure 1 you have 10-12 as habituation and 13th the training. Which is correct? Please make corrections.
Lines 476-477: “Less pronounced death occurred in CA2 and CA1 regions.”: There was no statistically significant difference at all in CA2.
There is no limitation section, and this study has a lot.
The comparison with the sham/control is not the same as comparison with control/TMT.
The only true result is the one in CA1 region.
The results of this study do not add to the translational research regarding the treatment of AD or neurodegenerative diseases. There is a benefit with regards to the pathophysiology of neurodegeneration.
Author Response
Response to Reviewer #5:
Thank you very much for your criticisms, they helped us improve the manuscript.
The is no translational or clinical hypothesis in the abstract regarding the results ie “The death of neurons in the CA1 field was significantly reduced in animals with combined use of memantine and VU 0422288.”, “However, the expression of genes characterizing neuroinflammation was markedly increased in the hippocampus of animals treated with memantine or VU 0422288 alone after TMT.”
We appreciate the possible interest from a translational point of view to our study. Our aim was an exploratory investigation on fundamental mechanisms of neurodegeneration in relation to application of a novel substance. For a conclusive translational outcome a larger study, focused on preclinical approach, would be required, which was not our goal in current work.
Line 60-61, “PAM of group III mGluRs”: Please, analyze and explain about the substance and its role. It is the first time mentioned here.
Thank you, revised.
Lines 125-128, “At the same time, the density of neurons in animals of the TMT+MEM and TMT+MEM+VU groups did not differ from that in the control group (Fig. 2B). The result indicates that the combined administration of memantine and VU0422288 is capable of improving the preservation of neurons in CA1 region of the hippocampus.”: Be very careful when jumping into results. In the figure there is difference between TMT and TMT+MEM+VU but not between TMT and TMT+MEM. Please rephrase the whole paragraph.
Thank you, we rephrased that.
Lines 135-136, “According to the results obtained, memory was impaired in all experimental animals, to the greatest extent – in the TMT group (Fig. 3).” Be very careful of the phraseology. The lowest p-value does not mean greatest extent. Means lowest probability of making a statistical error… Please rephrase.
Thank you, we rephrased that.
Method Section Lines 352-364: There is no power analysis regarding the number of the test subjects. There is also no reference regarding the dosage schema of the Memantine and the VU0422288. Why did not use injections through the whole experiment as it resembles more the clinical state (Patients with AD take Memantine for long periods of time).
We wanted to know if a short-term pharmacological suppression of excitotoxicity is able to tip the inflammatory balance towards resolution of neurodegeneration. Since it is not yet clear on a fundamental level, we consider our work to be some steps away from preclinical or therapeutic approaches. Larger groups would be necessary for a preclinical study, with current data gauging the necessary group size in relation to the effect size to achieve necessary power. At the moment, we suggest the η2 parameter for estimation of effect size, which is shown in figure captions.
Line 365 “animals were killed”: Replace with sacrifice. And also write the exact number of animals that were sacrificed by each method (decapitation and transcardiac perfusion) also in that part (3 PFA and 4 decapitation).
Thank you, fixed.
Lines 377-379 “On 11th through 13th days after TMT….”: In figure 1 you have 10-12 as habituation and 13th the training. Which is correct? Please make corrections.
Thank you, 10-12 is correct, fixed.
Lines 476-477: “Less pronounced death occurred in CA2 and CA1 regions.”: There was no statistically significant difference at all in CA2.
The CA2 region of TMT groups was not completely unaffected in TMT-treated groups - as seen from medians on the graph - but most cells maintained the appearance of a living neuron and were recognized as such by a segmentation model according to our criteria (medium intensity of staining of cytoplasm, visible nucleolus, no obvious ballooning or dehydration). Fixed in conclusion.
There is no limitation section, and this study has a lot.
We aimed to reduce the number of animals to a minimum
The comparison with the sham/control is not the same as comparison with control/TMT.
We administered sham injections of carrier saline (free of pharmacologically active agents) because it is appropriate for the control group to undergo equivalent amount of stress related to handling and procedures as the treatment groups.
The only true result is the one in CA1 region.
The results of this study do not add to the translational research regarding the treatment of AD or neurodegenerative diseases. There is a benefit with regards to the pathophysiology of neurodegeneration.
Reviewer 6 Report
I have appreciated how you built your experiment plan, and the rigorousness of your approach; unfortunately the results are far from being significant from a therapeutic perspective. While I also commend your writing style, I think that you should maybe elaborate a bit more in the Discussion about possible future applications of your current findings - do they recommend further actions, and if so, which ones; as it is now, I do believe that its relevance is limited for a readers' audience.
Author Response
Response to Reviewer #6:
I have appreciated how you built your experiment plan, and the rigorousness of your approach; unfortunately the results are far from being significant from a therapeutic perspective. While I also commend your writing style, I think that you should maybe elaborate a bit more in the Discussion about possible future applications of your current findings - do they recommend further actions, and if so, which ones; as it is now, I do believe that its relevance is limited for a readers' audience.
We appreciate the possible interest from a translational point of view to our study. Our aim was an exploratory investigation on fundamental mechanisms of neurodegeneration in relation to application of a novel substance. For a conclusive translational outcome, a larger study, focused on preclinical approach, would be required, which was not our goal in current work.
The revealed neuroprotective effect is seen in the increased preservation of CA1 neurons as a result of the combined therapy of memantine and group III PAM. This raises the question of the contribution of individual group III receptors. Of greatest interest for further research is the selective activation of the mGlu7 receptor subtype due to its distribution in the hippocampus and its enrichment in CA1 area.
Reviewer 7 Report
In this article from Pershina et al. the authors aim at studying the neuroprotective properties of the non-competitive NMDA receptor antagonist memantine, in combination with a positive allosteric modulator of metabotropic glutamate receptors of Group III, VU 0422288 on a TMT neurodegeneration model in rat.
The investigation is well written, the English language and style are fine, and the work well organized and significant. I thus recommend publication in International journal of Molecular Sciences after the following very minor revisions have been performed.
Caption in Figure 1 and 2 are not properly localized.
L396: subscript numbers in the molecular formula.
L398: minus sign must be inserted via the insert symbol command.
L402: Space after 8 (mm).
Reference section: apply the correct template for MDPI journals. See https://www.mdpi.com/authors/references. dots are mandatory for each journal abbreviations (e.g., Pharmacol. Res. and not Pharmacol Res
Author Response
Response to Reviewer #7:
Caption in Figure 1 and 2 are not properly localized.
Thank you, figures rearranged
L396: subscript numbers in the molecular formula.
Fixed.
L398: minus sign must be inserted via the insert symbol command.
Fixed.
L402: Space after 8 (mm).
Thank you, fixed.
Round 2
Reviewer 5 Report
Dear authors,
Whereas many of the comments were answered and corrected as instructed, some of them were not and the explanations were not, in my scientific opinion deemed appropriate.
"Method Section Lines 352-364: There is no power analysis regarding the number of the test subjects. There is also no reference regarding the dosage schema of the Memantine and the VU0422288. Why did not use injections through the whole experiment as it resembles more the clinical state (Patients with AD take Memantine for long periods of time).
We wanted to know if a short-term pharmacological suppression of excitotoxicity is able to tip the inflammatory balance towards resolution of neurodegeneration. Since it is not yet clear on a fundamental level, we consider our work to be some steps away from preclinical or therapeutic approaches. Larger groups would be necessary for a preclinical study, with current data gauging the necessary group size in relation to the effect size to achieve necessary power. At the moment, we suggest the η2 parameter for estimation of effect size, which is shown in figure captions."
You have deviated from the clinical practice that is already known and you do not even support it with reference that this proccess is already tested with positive results (your results are mostly negative).
"There is no limitation section, and this study has a lot.
We aimed to reduce the number of animals to a minimum"
The Limitation Paragraph does not exist.
Unfortunately, for these reasons, my decision is to reject this manuscript.
Author Response
Dear Reviewer,
In this work, we did not aim to completely stop the neurodegeneration caused by TMT. Such a task would indeed require a much more massive and prolonged pharmacological effort directed at the already well-described pathophysiological processes that develop in the hippocampus.
In our work, we undertook a short-term intervention in the development of neurodegeneration, which led to consequences that we observed even two weeks after the injections. If we had used memantine throughout the experiment, its action would have masked the additional effect of the mGlu receptor modulator. Therefore, despite the fact that the improvement in the state of the hippocampus is not as impressive as it can be achieved with a longer pharmacological treatment, we have shown effects of combined use of NMDA receptor antagonist (memantine) and presynaptic mGlu group III receptor PAM (VU0422288).
The aim of our work was to study the prospects of modulating the glutamatergic neurotransmission for the purpose of neuroprotection. The usual targets for the modulation of glutamatergic transmission are ionotropic glutamate receptors, glutamate and calcium transporters. The potence of ligands of metabotropic receptors is less known. Unfortunately, clinical application of those ligands remains limited (Witkin et al, 2022 doi: 10.1016/j. pbb.2022.173446). However we believe that the fine modulation of glutamatergic synaptic transmission using mGluR ligands remains promising. The results obtained in our work confirm this point of view.
We have added a limitations section to the manuscript and also updated references on substances.
Thanks again for your interest in our work.
Round 3
Reviewer 5 Report
Dear authors,
Thank you for taking into consideration the comments.